# Media Influence on the Perceived Safety of Dietary Supplements for Children: A Content Analysis of Spanish News Outlets

**DOI:** 10.3390/nu17060951

**Published:** 2025-03-08

**Authors:** Rosa Melero-Bolaños, Belén Gutiérrez-Villar, Maria Jose Montero-Simo, Rafael A. Araque-Padilla, Cristian M. Olarte-Sánchez

**Affiliations:** 1Department of Management, Universidad Loyola Andalucía, 14004 Córdoba, Spain; belengut@uloyola.es (B.G.-V.); jmontero@uloyola.es (M.J.M.-S.); raraque@uloyola.es (R.A.A.-P.); 2Department of Psychology, Universidad Loyola Andalucía, 41704 Sevilla, Spain

**Keywords:** dietary supplement, children, media, health

## Abstract

**Background/Objectives:** The influence of media on the public opinion, especially regarding health topics, is profound. This study investigates how Spanish media may reinforce a positive image of dietary supplements for children, potentially leading to harmful health attitudes and behaviors. **Methods:** The researchers conducted a quantitative content analysis of 912 news articles from Spanish media outlets discussing dietary supplements for children between 2015 and 2021. They used a frequency analysis and a proportion comparison to analyze variables such as the reach of news, tone of news, mentions of health professional consultation, association with natural products, media specialization, intertextuality, and headline mentions. **Results:** The study found a 60% increase in publications discussing dietary supplements for children during the study period. The content analysis indicates that these articles predominantly present dietary supplements in a positive light, often without robust scientific evidence. Furthermore, many do not emphasize the need for medical consultation, which may contribute to unsupervised consumption, particularly among minors. This highlights the critical importance of professional guidance when considering dietary supplements for children. Additionally, the frequent emphasis on the “natural” attributes of these products raises concerns regarding consumer perceptions and potential safety risks. **Conclusions:** The study reveals a problem regarding the portrayal of dietary supplements for children in Spanish media. The overly optimistic image, lack of scientific basis, and failure to recommend medical supervision may contribute to unsupervised consumption among minors, risking their health due to misinformed decisions influenced by media portrayal.

## 1. Introduction

Various terms can describe dietary supplements (DSs), such as nutritional supplements, dietary supplements, or plant-based compounds. DSs are defined in Directive 2002/46/CE of the European Parliament [1] as products intended to supplement the regular diet, consisting of concentrated sources of nutrients or other substances with a nutritional or physiological effect, in a simple or combined form, marketed in a dose form, i.e., capsules, pills, tablets, and other similar forms of liquids and powders that are to be taken in small unit quantities.

In recent years, there has been a substantial increase in the consumption of DSs across Europe, even more during and after the COVID-19 pandemic, where the population used them to strengthen the immune system and thereby protect against the coronavirus [2]. Vitamins, minerals, Omega-3 and probiotics are among the most popular [3]. Around 20% of European consumers are estimated to acknowledge taking some supplements [4], although it is not always motivated by real nutritional needs [5]. These figures highlight the high consumer confidence; a total of 70% believe in their safety and quality [3]. However, several systematic reviews and meta-analyses have shown that most DS are ineffective in preventing or treating diseases [6]. Many of them can be harmful [7], particularly to people with specific diseases [8], with serious safety issues being reported due to their interaction with other medications [9].

In children, DSs are popular globally, not just for addressing nutritional deficits but also for the general health enhancement [10,11]. Despite the rising use of DSs in children, there is a lack of knowledge about their properties, risks, and drug interactions [12]. Parents use DSs for children’s nutritional deficits [10], to protect against infections [13], or to improve sleep [14], frequently without solid evidence supporting their effectiveness [15]. DSs are also used to improve the symptoms of Autism Spectrum Disorders (ASD) and other behavioral problems [16]. However, their effectiveness is questioned [17]. Furthermore, studies show that DSs can be more harmful in children due to their developing bodies; therefore, the effects they might have in adult life are unknown [7,18].

With the increasing ease of acquiring dietary supplements in various establishments, from pharmacies and specialty stores to supermarkets, cosmetic stores, and even markets, it is increasingly important to educate the public to help them make healthier decisions [19]. This wide availability requires clear guidance on its appropriate and safe use. At this point, it is worth asking about the primary sources of information or the advice that has led the parents to purchase DSs for their children. Generally, the media emerges as one of the most significant external sources for obtaining information on specific health topics and healthy habits [20]. The media influence is combined with other factors, such as recommendations from family and friends or from healthcare providers, in the case of DSs [21]. This influence is specified in variables such as the intention to purchase based on one’s perception of DSs [22].

The influence that the media has on creating or reinforcing attitudes is well-known across all public opinion topics. They are a source that people use to inform themselves and generate opinions on various issues [23]. Many of our ideas are based on an image constructed from what is said in the media. One of the media’s functions is to provide the public with sufficient information and knowledge to encourage critical thinking and help them make decisions that directly affect their lives.

The influence of media on public perception of dietary supplements can be better understood through the lens of the following two prominent media theories: the Agenda-Setting Theory and the Framing Theory. The Agenda-Setting Theory, developed by McCombs and Shaw, posits that media plays a crucial role in determining which issues the public considers essential [24]. By emphasizing specific topics, such as the benefits of DSs, the media can influence the salience of these issues in the public mind. The Framing Theory suggests that the media tells us what to think about and how to think about it [25]. Media can promote interpretations and evaluations of DS efficacy and safety by selecting certain aspects of DS use and making them more salient [26]. These theories provide a solid foundation for understanding how media coverage can shape the public opinion on DS use, particularly in children, and justify the importance of analyzing media content in this context [27]. We must consider that a positive image projected to the public could induce counterproductive or harmful attitudes and behaviors toward health [16,28].

However, as far as we know from the reviewed literature, how the supplements for children are portrayed in media is still a rarely studied question. So, more evidence is needed to analyze and compare how the media in different countries portray supplements for children, particularly in regulatory contexts with varying advertising restrictions.

This study seeks to fill these gaps by analyzing the media coverage on DSs for children in Spain. This analysis will help us better understand how the news appearing in the media could reinforce a positive image. To achieve this objective, we have compiled the news items written in the Spanish media, broadcasted between 2015 and 2021, where DS use in children is discussed. Compared to other European countries, Spain has positioned itself as one of the leading markets for dietary supplements, ranking fifth in revenue, behind only Italy, Germany, the United Kingdom, and France [29]. This positioning reinforces Spain’s role in the health and wellness industry at the European level. In 2023, the Spanish dietary supplement market experienced a 6.3% increase compared to the previous year, reaching a revenue of €1.849 billion in pharmacies and parapharmacies. Additionally, the market has maintained an annual growth rate of 4.4% over the past five years.

## 2. Materials and Methods

Based on the general objective presented above, the following specific objectives were defined:To understand the coverage of DSs for children in the Spanish media.To ascertain how a positive image of DSs for children is projected in the Spanish press.

The research method applied, a quantitative content analysis, involves systematically categorizing and statistically analyzing communication content to determine the frequency of certain elements. This technique is broadly employed in various health research contexts, demonstrating its effectiveness and adaptability for drawing insights from diverse informational sources.

### 2.1. Sample Selection

A poly-staged sampling procedure was used to answer these research questions. Firstly, a period was established, specifically news published in the Spanish media between 2015 and 2021, to give the sample a particular temporal perspective. The news were located through the specialized search engine “MyNews” [30], an electronic resource allowing the users to download content queries published in media through an advanced search engine with rules and multiple filtering options. Therefore, to focus our search on topics relevant to child nutrition, the following rules were established: search for articles containing (foods OR supplements) AND (nutritional OR dietetic) AND (infant OR children OR pediatric). Regarding the search filters, the following were applied:Media Coverage: all media were selected, including local, regional, national, and international.Territory: the filters were adjusted to include news from all over Spain, ensuring a comprehensive view of national coverage.News Genre: all news genres were included, and the search was not limited to health and wellness sections, so that a broader perspective on the media representation of the topics is studied.Type of Media: the search was extended to all available in MyNews, including news agencies, digital media, print media and magazines.Position in the Article: the rule was applied to the entire content of the articles, not only to titles and subtitles.Media Sections: all sections available in the media were selected without restricting them to specific categories.Finally, the data range collected included news from 1 January 2015, to 31 December 2021.

Throughout the sorting process of the news collected through “MyNews”, our research team also performed manual filtering. At this stage, we excluded articles not directly related to our study, such as those that addressed supplements from a humanitarian aid perspective or were marked as “not suitable for children”. This simultaneous sorting and filtering methodology ensured the relevance and accuracy of the content included in our food and nutritional supplements analysis in child populations.

It was also the research team itself that classified the media as generalist or specialized. This classification was based on the influence that the specialized media have on the perceived credibility of the content. Generalist media address various topics and usually aim at a more general audience. These media tend to cover news and current events from a broader and less detailed perspective. On the other hand, specialized media focus on specific areas of knowledge, in this case, health or nutrition, and are aimed at audiences looking for more detailed and technical information. Due to their in-depth, expert approach, these media are perceived as more credible, especially when disseminating complex knowledge. This resulted in a database of 912 relevant news items.

### 2.2. Study Variables

Below, we discuss the variables analyzed concerning the proposed objectives (Table 1).

#### 2.2.1. Reach of News About DS for Children in the Media

The media grants or denies a topic the status of newsworthiness. Thus, agenda setting is the process followed by the media to select and determine the events that will become news. When the number of messages about a specific topic increases over time, the public perceives the topic as more relevant [31]. This work analyses the evolution of the quantity of news about DSs for children.

Another critical aspect in evaluating the news’ reach is media’s digitalization. In this regard, and since 2021, the Internet has been the most used medium in Spain, with an 87.6% penetration rate, compared to the press at 13.7% [32]. This study assesses the prevalence of digital media coverage on DSs for children.

#### 2.2.2. Tone of the News

The tone of the news was analyzed to assess the image projected in the news about DSs for children. The following four stances were identified: skeptical, merely descriptive, presenting multiple perspectives, and overall supportive [33].

News with a skeptical tone show opinions against DS consumption or highlight the risks over the benefits. As an example of projecting a skeptical image, we found comments in the news like the ones presented below:


*Dietary supplements are not miracle pills. Few claims are supported by science, and many others have proven ineffective.*



*First, you should know that no dietary or vitamin supplements prevent, treat, or cure coronavirus infection.*


As for the news where the image of DSs for children is more neutral, with a merely descriptive tone, we found texts such as the following:


*Provide comprehensive consumer information with specific labelling of food supplements so that a medicinal plant cannot be sold as a supplement and a medicinal product at the same time. Establish maximum and minimum limits. This way, reference values will be well-defined, especially for children.*



*The sale of foods, beverages, and supplements capable of boosting the immune system has skyrocketed when consumers increasingly take preventive measures against the virus.*


For news with a tone showing multiple perspectives on the image of DSs for children, we found texts that incorporated both benefits and risks, as seen in the examples below:


*Some studies suggest that supplementing Omega 3 could help prevent or reduce the symptoms of some pathologies (…). If we have a balanced diet, it is rare to have a deficit of any of these acids unless there is an absorption problem. Therefore, a priori supplementing the diet would not be necessary unless the doctor subscribes.*



*When a child is fully fed, he or she does not usually need vitamin supplements since most foods are full of proteins, nutrients, vitamins, and everything needed to grow healthy and strong (…). However, as we say, there may be times when your body needs a “little help”… The important thing is never to decide on your initiative regarding the type of supplement and dosage, but always follow the pediatrician’s instructions: some substances, if taken in excessive amounts, can cause the child to suffer side effects.*


Finally, among the news items, we found positive images of DSs for children, where more emphasis is placed on their benefits, representing general support for their consumption, with statements such as:


*They are ideal for children because of their fruity flavor and soft texture. The good thing about this product is that it is easily assimilated, helps reduce fatigue, and improves the immune system’s functioning.*



*Both the American Academy of Pediatrics (AAP) and the Nutrition Committee of the Spanish Association of Pediatrics recommend supplementing infants under one year of age with vitamin D daily from birth and those older than this age and adolescents who, due to their diet, do not ingest the recommended daily requirements. No studies in children suggest what level of sun exposure is necessary to dispense with dietary supplements.*


#### 2.2.3. Appeal to Consult with Healthcare Professionals

As recommended by health authorities [8,34], the study checked if the articles recommended consulting health professionals (doctors, pediatricians, specialists, professionals, dietitian-nutritionists) before using DSs. This advisory presence in the news moderates the impact of a positive image.

#### 2.2.4. Association of DS with Natural Products

The positive image of DSs is reinforced by the belief that they are natural products and, therefore, are perceived as safe. The natural origin of many of these products leads to a sense of safety [35]. Hence, the mention of the ingredients or the natural character of DSs in the news reinforces their positive image.

#### 2.2.5. Media Specialization and Credibility

Regarding credibility, specialized journalism is deemed more credible due to its in-depth knowledge and, therefore, greater accuracy on the topic it deals with, an issue that is accentuated when the journalist assumes the task of disseminating complex knowledge [36]. Thus, this study compares the credibility of specialized versus generalist sources in reporting on DSs for children in the media.

#### 2.2.6. Intertextuality and Reliance on Information

Intertextuality is related to mixing statements and texts or deriving one text from another. Intertextuality aims to give more objectivity to a news item [37], generating more trust in the reader. This study considers whether the journalistic texts cite documentaries (journals, books, reports) and personal sources (doctors, scientists, research centers, etc.) or both, compared to those that do not mention sources.

#### 2.2.7. Mention of DS in the Headline

This variable is essential because headlines are the most visible element of the news [20] and often the only part read by many people [38], with 49.3% of Spaniards reading just headlines [39].

### 2.3. News Codification Procedure and Techniques of Analysis

The 912 news articles were entered into an Excel spreadsheet and coded for the discussed variables. Four researchers independently coded a random sample of 20 articles to ensure coding accuracy. After revising the coding criteria due to initial disagreements, especially on ‘tone’, a second reliability check was conducted with 23 new articles. We used Krippendorff’s Alpha for intercoder reliability [40], achieving values above 0.8 for all variables.

This study used qualitative variables, analyzing them through frequency and proportion comparisons. The “Tone of the News” variable was split into “overall supportive” and others. While acknowledging that dichotomizing an originally polytomous variable may result in some information loss, our methodological decision to bifurcate “Tone of the News” categories into “Overall supporting” and “Other Tones” was based on carefully evaluating its merits and limitations. Although potentially simplifying the inherent complexity of tone variability in the news, this approach offers several advantages that align with our research objectives and enhance the study’s overall robustness. Primarily, it allows for a more focused analysis of positive-tone news, which is central to our research aims.

Furthermore, this dichotomization yields larger sample sizes for each group, thereby increasing statistical power and enhancing the reliability of our results. The resulting binary classification also facilitates a more precise interpretation and effective communication of findings, particularly to a broader audience. Moreover, in many practical contexts, this dichotomous distinction between the clearly positive news and others may prove more actionable and valuable than a more granular categorization.

So, we compared these categories’ communicative strengths using the Z-test, which evaluates the presence/absence of variables such as professional consultation, association with natural products, media specialization, intertextuality, and headline mentions.

The Z-test for proportion comparisons is a robust statistical tool for analyzing categorical or qualitative variables in large-sample studies. This method excels in determining significant differences between the proportions of two independent groups, making it invaluable for comparing distinct strategies or treatments [41]. The Z-test’s strength lies in its capacity to provide meaningful insights into group differences while maintaining statistical rigor, especially when dealing with substantial sample sizes that ensure the validity of its underlying normal distribution approximation [42].

## 3. Results

The research analyzed dietary supplement (DS) for children coverage in Spanish media, identifying 912 news items over seven years, with an average of 130 items annually. There was a significant 60% increase in DS-related publications from 2015 to 2021. Additionally, the study highlighted a shift towards digital media, where online sources grew from accounting for 70.5% of such news in 2015 to 85.7% by 2021. These trends are depicted in Figure 1.

Analyzing the frequencies of variables (Table 2), most news are broadcast through generalist media (80%), reaching a broader audience. About 72.7% of news reporters avoid linking DSs with natural products, and the mention of DSs in the headline appears in 43% of cases, though other topics, like benefits or health problems, are more common. Most news (85%) use specialized sources for DS information, yet only 27% recommend consulting a specialist, suggesting a reliance on intertextuality or underestimating the consequences of DS use.

The second objective assessed how positively the DSs for children are portrayed in Spanish media. An analysis of the four identified tones revealed a dominant overall supportive stance at 57%, skeptical at 17%, descriptive at 2%, and multiple perspectives at 24%.

As mentioned in Section 2.3, “Tone of News” has been dichotomized for proportion tests. So, the “Tone of the News” variable was split into “Overall Supportive” (57%) and “Other Tones” (43%). So, the news projecting a positive image of DSs for children were analyzed for proportional differences against other variables listed in the theoretical basis (Table 3).

Firstly, we compared the association between an “Overall Supportive” tone and “Reference to a natural product”, finding statistically significant differences (*p* = 0.000). News linking DSs with natural products more often had a positive tone (65.5%) compared to those that did not (53.4%).

Differences also emerged in the media used for publication (*p* = 0.000). It is more likely to find news with an “Overall Supportive” tone in specialized print media (69%).

Similar results were found in intertextuality comparisons, showing significant differences (*p* = 0.000), where the news lacking personal or documentary sources had a more Overall Supportive tone (75% vs. 54% “Other Tones”).

On the other hand, the differences in the proportions between the presence of “Appeal to consult a healthcare professional” and “DS mentioned in the headline” in the Tone are also statistically significant, although in a different direction to the previous variables mentioned, as “Appeal to consult a healthcare professional” is more prevalent in the news with “Other Tones” (61.0%). Similarly, the “Appeal to consult a healthcare professional” is more likely to appear in the news with “Other Tones” (42.4%).

## 4. Conclusions

This study demonstrates that the coverage of DSs for children in the Spanish media has increased by 60% over seven years, potentially raising consumer awareness due to increased media prevalence [31]. Post-COVID-19, DS coverage surged in digital media, while slightly declining in offline media. This shift has broadened the reach, particularly among the parents who prefer digital media for information, contrasting with the older audience of offline press [43]. Importantly, 80% of DS information is disseminated through generalist media, reaching a broader audience.

Regarding the media’s portrayal of DSs for children, we observed a positive image that confirms its suitability to public opinion [23]. According to Scheufele and Tewksbury [25], this phenomenon is attributable to media framing, which often highlights the benefits of DSs without adequately discussing the risks or the need for medical supervision. This tendency enhances the visibility and interest in DSs and may promote a feeling of necessity and efficacy, not always supported by robust scientific evidence. Such framing may lead parents and caregivers to make potentially ill-informed decisions, contrary to specialized medical recommendations.

Furthermore, the scarce discussion in the media about the risks associated with DS consumption may contribute to a lower perception of risk among the public. It is particularly worrying in the context of children’s health, where supplementation decisions should be made with caution and based on professional advice.

Our findings align with the trends observed in other studies on health communication. For example, studies such as those by Weeks [44] and Milazzo [45] have also reported a growth in media attention towards alternative health and supplementation issues, finding a predominantly positive tone in the media coverage of alternative and complementary therapies. As in our study, while the coverage is primarily positive, the criticisms regarding the effectiveness and safety of these interventions remain underrepresented, which resonates with the observations of Caulfield et al. [33] on vitamin D supplementation, where the promotion of supplementation in the media without the adequate discussion of its potential risks was highlighted.

A recent study [16] indicates that more than 75% of Spanish pediatricians recommend dietary supplements. This high recommendation rate could be reflected in the generally positive tone found in our analysis of news articles, where 57% were “Generally in favor” of supplements. The frequent use of these products in daily pediatric practice may coincide with the positive media coverage observed in our study. This observation suggests that the positive representation of dietary supplements for children in the media could be influenced, in part, by standard prescribing practices among health professionals. However, it is relevant to consider that this correlation does not necessarily imply causation and that other factors could contribute to the supportive tone in the news coverage.

In this regard, news items showing a more positive stance often lacked documentary or personal sources to substantiate claims, warranting cautious interpretation. This is concerning as such media might promote economically driven information, especially significant given the recent sales increase in these products [2]. This finding is consistent with the concerns expressed in the medical article [46] regarding the lack of child-specific evidence for many nutritional recommendations targeting young athletes.

The data show that 73% of positive news about children’s DSs does not encourage consulting specialists. This trend supports findings that DS consumption often occurs without medical advice [3,13].

The news associating DSs with natural products have a higher proportion of positive tones (65.5%), enhancing their appeal due to the common belief that “natural” implies safety. This perception is more potent in preventive than curative contexts [47]. However, “natural” is a vague term, raising concerns about its safety and highlighting the need for evidence-based evaluations [48].

Interestingly, news in specialized media is less frequent but often more positive about DS for children. Specialized outlets are more reliable because they provide detailed information on specific topics, unlike generalist media, which may cover topics superficially [49]. This perception enhances the favorable view of DSs in specialized publications.

Omitting “supplement” from supportive news headlines may increase persuasion by focusing on the benefits rather than the product. Research in this field suggests that news headlines, designed to capture the reader’s attention on a psychological level, can play a crucial role in how people perceive and interact with it. Furthermore, research from the Centre for Media Engagement highlights significant evolution in headlines with the shift from traditional to digital media, emphasizing the strategic role of headlines as demonstrated by the common practice of A/B testing headlines in real time [50].

The media’s increased and predominantly positive portrayal of dietary supplements (DSs) correlates with a higher consumption. However, their effectiveness is debatable, as studies show that DSs may be ineffective or harmful [6]. It creates several problems; one is related to guiding the consumers to purchase products that could be ineffective, as there is not enough evidence to prove their effectiveness. On the other hand, it creates a health problem, since it could harm consumers by not being subjected to the same control as other health products [7,9]. It is important to remember that DSs can be beneficial in specific cases, were their effects cannot be achieved due to a diet or some deficiency or pathology. Therefore, it should be contraindicated for children to consume DSs when there is no precise diagnosis of these deficiencies.

Despite the intriguing findings of our research, it is crucial to consider the certain limitations. Specifically, this study does not set out to measure the impact of the analyzed content on the readers’ perceptions or behaviors. Instead, we have analyzed whether there may be reasons for concern based on the image of DSs conveyed by the media. The study’s ability to determine how headline exposure may modify the individuals’ opinions, attitudes, or actions, is limited. Therefore, studies like the one presented here should be complemented by others exploring the influence of such content on reader behavior.

Images influence how the news is interpreted and percieved, as they can evoke emotions, reinforce certain viewpoints, and contextualize the information. The choice of images can affect the reader’s attention, information retention, and the perceived credibility of the news, so it could be beneficial for future research to include the analysis of images along with the text to gain a more comprehensive understanding. Another aspect to consider is that our study did not specifically identify the section in which the analyzed news was published. Given that not all sections of a media outlet attract the same number of readers or generate the same level of credibility, the lack of this differentiation could influence the interpretation of the results. Analyzing the audiences and profiles of those who access news in different media would allow us to determine how the information about DSs for children is distributed and received in various media contexts. Understanding who the readers or viewers are and how they access the news will help us identify possible key differences in the exposure and interpretation of the content. This information is crucial to developing communication strategies that are effective and relevant to different segments of the population, ensuring that messages about children’s health are adequately received and understood by various groups. 

Likewise, to eliminate the possible biases in evaluating the emotional tone of the news about DSs, it would be advisable in the future research to take advantage of the advances in the field of artificial intelligence and sensory evaluation techniques. In this sense, the use of new techniques such as the Natural Language Process (NLP), which would allow key information to be extracted from texts and detect emotions or feelings in comments, opinions, or social networks, or AI-driven sentiment analysis, would strengthen the evidence obtained [51,52].

Acknowledging that media reporting does not occur in a vacuum is essential. Commercial interests and potential conflicts of interest can significantly shape how dietary supplements are portrayed. To address this potential influence, future research could analyze media content sources, examine the relationship between regulatory changes and shifts in media portrayal of dietary supplements, and investigate how different stakeholders influence media narratives around children’s dietary supplements. By incorporating these factors, we can provide a more comprehensive understanding of how the media shapes perceptions of dietary supplements for children, acknowledging the complex interplay of commercial interests, regulatory frameworks, and public health concerns.

The study provides some recommendations for media managers and policymakers. Given the risks of uncontrolled consumption of DS, the media should promote and take greater care of intertextuality by using more credible sources. Contrasting the sources used and making them explicit can help evaluate their veracity and avoid leading people to make erroneous decisions. News about DSs should consistently emphasize the need for parents and caregivers to consult healthcare providers before administering dietary supplements to children. It could be reinforced through expert quotes or case studies demonstrating the consequences of unsupervised supplement use. Journalists also should critically assess and challenge the widespread notion that “natural” products are inherently safe. This narrative can mislead consumers into thinking that supplements, due to their natural origin, are free from risks. Clear communication about the difference between “natural” and “safe” could help curb misconceptions.

On the other hand, the public administration should monitor compliance with the regulations, ensuring that the news does not have any commercial interests that aren’t publicly known. Policymakers should fund research into the dietary supplements’ long-term effects and efficacy, particularly for children, to inform evidence-based health policies. It could include studies on the potential interactions between supplements and medications commonly used by children. Government-led campaigns should focus on educating the public about the safe use of dietary supplements, emphasizing the importance of consulting healthcare professionals before using such products. These campaigns could follow successful strategies in food literacy with a multi-component approach, combining theoretical sessions with practical and interactive activities and relying on digital tools [53].

The growing use of nutritional supplements by more people, alongside the often favorable media portrayals, underscores the need to monitor their effects on diets, health, and disease prevalence. Public policymakers must analyze the consumption of these products in detail, especially in children, and clarify their possible role in preventing and treating diseases, thus balancing public perception, influenced by the media, with scientific evidence and public health considerations.

## Figures and Tables

**Figure 1 nutrients-17-00951-f001:**
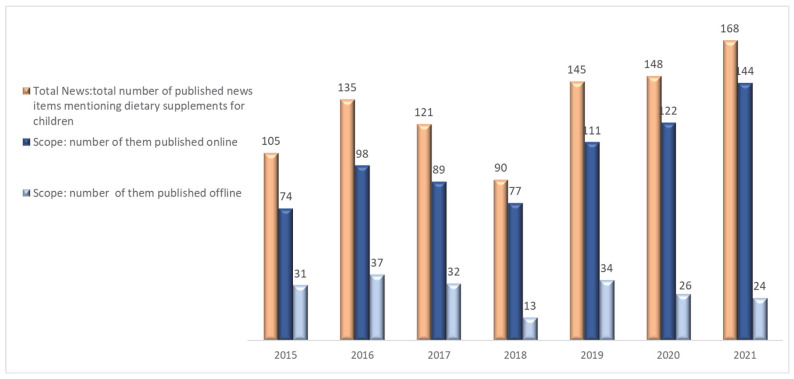
Annual evolution of the number of news items published about DSs in online and offline media.

**Table 1 nutrients-17-00951-t001:** Variables of the study.

Variables	Categories
Reach	Number of news
Tone of news	Skeptical Merely descriptive Multiple perspectives Overall supportive
Appeal to consult a healthcare professional	Yes No
Association with natural products	Yes No
Specialization	Specialized in healthcare media Generalist media
Intertextuality	Documentary sources Personal sources Both sources No sources
Mention of DSs in headline	Yes No

**Table 2 nutrients-17-00951-t002:** Frequency of variables.

Variables	Number of News (%)
**Tone of news**	
Skeptical	155 (17%)
Merely descriptive	18 (2%)
Multiple perspectives	221 (24%)
Overall supportive	518 (57%)
**Tone of news dichotomized**	
Overall supportive	518 (57%)
Other tones	394 (43%)
**Appeal to consult a healthcare professional**	
Yes	243 (27%)
No	669 (73%)
**Association with natural products**	
Yes	258 (28.3%)
No	654 (71.7%)
**Type of media**	
Specialized in Healthcare	184 (20%)
Generalist	728 (80%)
**Intertextuality**	
Documentary sources	102 (11%)
Personal sources	403 (44%)
Both sources	273 (30%)
No sources	134 (15%)
**Intertextuality dichotomized**	
Use specialized sources	778 (85%)
No sources	134 (15%)
**Mention of DS in the headline**	
Yes	388 (43%)
No	524 (57%)

**Table 3 nutrients-17-00951-t003:** The contrast of proportions (Z-test) between the tone of news and other categories of variables reinforcing the positive image of DSs.

Type of News That Reinforces a Positive Image of DSs	Tone of News		
	Overall Supportive	Other Tones	*p*-Value
Appealing to consulting a healthcare professional	14.7%	42.4%	0.000
Linking DSs with natural products	65.5%	53.4%	0.000
Appearing in specialized healthcare media	69.0%	53.0%	0.000
Using specialized sources	54.0%	75.0%	0.000
Mentioning DS in the headline	51.0%	61.0%	0.003

## Data Availability

The dataset is available on request from the authors because the data source is not freely accessible.

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
