# Peer review of "Media Influence on the Perceived Safety of Dietary Supplements for Children: A Content Analysis of Spanish News Outlets"

_nutrients, 2025, doi:10.3390/nu17060951_

Round 1
Reviewer 1 Report
Comments and Suggestions for Authors
I would like to thank for the opportunity to review this manuscript. Overall, this manuscript is well written. Please see the following comments to consider to further increase the quality of this manuscript.
While the introduction provides a strong background on dietary supplement use and media influence, it lacks a direct articulation of the specific gap this study addresses. Clearly state what previous studies have not examined regarding the media’s role in shaping public perceptions of dietary supplements for children.
Highlight why a Spanish media focus is particularly relevant.
The study could benefit from a more explicit theoretical underpinning, such as Agenda-Setting Theory or Framing Theory, to justify the role of media in shaping public opinion. Introduce media theories to contextualize the study’s hypotheses.
The manuscript mentions using MyNews to gather articles but does not provide a detailed explanation of search string formulation or criteria for inclusion/exclusion beyond humanitarian aid references.
Were regional or national outlets included? Did the study account for media bias or political affiliations? Provide more details on search parameters to ensure replicability.
The study applies Z-tests for proportion comparisons, but further explanation is needed on the rationale for choosing this test over other statistical approaches.
How were multiple comparisons controlled for? Include a justification for the statistical choices and, if applicable, whether corrections (e.g., Bonferroni) were used.
Ensure that tables and figures are self-explanatory without requiring extensive text reference.
The study currently classifies media as generalist vs. specialized, but further categorization (e.g., newspapers, blogs, social media, television) could provide deeper insights. A breakdown by media format and audience demographics.
While the study references prior research, it does not sufficiently compare its findings to existing work on media framing of health topics. Discuss how results align or diverge from previous studies on dietary supplement communication.
The study assumes a direct influence of media on consumer perceptions without considering pre-existing biases (e.g., industry-sponsored media content, government health advisories). Introduce a discussion on commercial interests and potential conflicts of interest in media reporting.
The conclusion hints at policy recommendations, but these should be more specific. Provide explicit recommendations for policymakers and journalists on improving public health communication regarding dietary supplements.
More innovative suggestions for future research are recommended. For example, in future research one could use natural language processing (NLP) and AI-driven sentiment analysis to assess bias and emotional tone in media reports on dietary supplements. Please see a recent research by Ravšelj et al., (2025) that might be useful by adding the most recent insights from the AI research.
Ravšelj D, Keržič D, Tomaževič N, Umek L, Brezovar N, A. Iahad N, et al. (2025). Higher education students’ perceptions of ChatGPT: A global study of early reactions. PLoS ONE 20(2): e0315011. https://doi.org/10.1371/journal.pone.0315011
Author Response
Coment 1: I would like to thank for the opportunity to review this manuscript. Overall, this manuscript is well written. Please see the following comments to consider to further increase the quality of this manuscript.
Response 1: We would like to thank the reviewer for their valuable comments and for the opportunity to review this manuscript. We appreciate their positive evaluation of the work and the suggestions for improving its quality. In pdf, we have highlighted our replies in blue and the changes incorporated into the manuscript in red.
Coment 2: While the introduction provides a strong background on dietary supplement use and media influence, it lacks a direct articulation of the specific gap this study addresses. Clearly state what previous studies have not examined regarding the media’s role in shaping public perceptions of dietary supplements for children.
Highlight why a Spanish media focus is particularly relevant.
Response 2: We appreciate your observation. We have tried to make clearer what the gap of literature is in the text and why a Spanish media focus could be relevant.
Lines 98-112
However, as far as we know in reviewing literature, how supplements for children are portrayed in media is still a less studied question. So, more evidence is needed to analyze and compare how the media in different countries portray supplements for children, particularly in regulatory contexts with varying advertising restrictions.
This study seeks to fill these gaps by analyzing media coverage on DS for children in Spain. This analysis will help us better understand how the news appearing in the media could reinforce a positive image. To achieve this objective, we have compiled the news items written in the Spanish media, which appeared between 2015 and 2021, where DS for children are discussed. Compared to other European countries, Spain has positioned itself as one of the leading markets for dietary supplements, ranking fifth in revenue, behind only Italy, Germany, the United Kingdom, and France [29]. This positioning reinforces Spain's role in the health and wellness industry at the European level. In 2023, the Spanish dietary supplements market experienced a 6.3% increase compared to the previous year, reaching a revenue of €1.849 billion in pharmacies and parapharmacies. Additionally, the market has maintained an annual growth rate of 4.4% over the past five years.
Coment 3: The study could benefit from a more explicit theoretical underpinning, such as Agenda-Setting Theory or Framing Theory, to justify the role of media in shaping public opinion. Introduce media theories to contextualize the study’s hypotheses.
Response 3: We think that a better theoretical context could improve the paper, following your comments, so, we have introduced references and theories in the text, as follows:
Lines 85-95
The influence of media on public perception of dietary supplements can be better understood through the lens of two prominent media theories: Agenda-Setting Theory and Framing Theory. Agenda-Setting Theory, developed by McCombs and Shaw, posits that media plays a crucial role in determining which issues the public considers essential [24]. By emphasizing specific topics, such as the benefits of DS, the media can influence the salience of these issues in the public mind. Framing Theory suggests that the media tells us what to think about and how to think about it [25]. Media frames can promote interpretations and evaluations of DS efficacy and safety by selecting certain aspects of DS use and making them more salient [26]. These theories provide a solid foundation for understanding how media coverage can shape public opinion on DS use, particularly for children, and justify the importance of analyzing media content in this context [27].
Coment 4: The manuscript mentions using MyNews to gather articles but does not provide a detailed explanation of search string formulation or criteria for inclusion/exclusion beyond humanitarian aid references.
Were regional or national outlets included? Did the study account for media bias or political affiliations? Provide more details on search parameters to ensure replicability.
Response 4: We thank the reviewer for his valuable comments and suggestions regarding the use of the MyNews database and the formulation of our search methodology. In response to his questions, we have provided additional details on the search parameters used, with the aim of improving the clarity and replicability of our study. Following his guidance, we have comprehensively described these parameters in section 2.1 of the reviewed article. This increased transparency ensures that other researchers can accurately understand and replicate our methodology.
Regarding your last question, We would like to highlight that, although we are aware that media bias and political leanings can influence the coverage of certain topics, our main objective was to understand the coverage of dietary supplements for children in the Spanish media and to determine to what extent a positive image of children's dietary supplements is projected in these media. To achieve this, we applied a quantitative content analysis method, which involves systematic categorization and statistical analysis of the content of communication. By focusing on specific and measurable variables, our study provides a framework for analyzing the coverage of dietary supplements in the Spanish media, regardless of their editorial or political leanings. This methodological approach allows us to draw data-driven conclusions thus contributing to a clearer and more balanced understanding of how children's dietary supplements are presented in the media.
The specific modifications we have made to that section are presented below:
Lines 127-163
The news was located through the specialized search engine "MyNews" [30], an electronic resource allowing users to download content queries published in media through an advanced search engine with rules and multiple filtering options. So, to focus our search on topics relevant to child nutrition, the following rules were established: search for articles containing (food ORsupplements) AND (nutritional OR dietetic) AND (infant OR children OR pediatric). Regarding the search filters, the following were applied:
- Media Coverage: All media were selected, including local, regional, national, and international.
- Territory: The filters were adjusted to include news from all over Spain, ensuring a comprehensive view of national coverage.
- News Genre: All news genres were included, and the search was not limited to health and wellness sections. So, a broader perspective on the media representation of the topics is studied.
- Type of Media: The search was extended to all available in MyNews, including news agencies, digital media, print media and magazines.
- Regarding Position in the Article, the rule was applied to the entire content of the articles, not only to titles and subtitles.
- Regarding Media Sections: All those available in the media were selected without restricting them to specific categories.
- Finally, the data range collected included news from January 1, 2015, to December 31, 2021.
Throughout the sorting process of the news collected through "MyNews", our research team also performed manual filtering. At this stage, we excluded articles not directly related to our study, such as those that addressed supplements from a humanitarian aid perspective or were marked as "not suitable for children." This simultaneous sorting and filtering methodology ensured the relevance and accuracy of the content included in our food and nutritional supplements analysis in child populations.
It was also the research team itself that classified the media as generalist or specialized. This classification was based on the influence that specialized media have on the perceived credibility of the content. Generalist media address various topics and usually aim at a more general audience. These media tend to cover news and current events from a broader and less detailed perspective. On the other hand, specialized media focus on specific areas of knowledge, in this case, health or nutrition, and are aimed at audiences looking for more detailed and technical information. Due to their in-depth and expert approach, these media are perceived as more credible, especially when disseminating complex knowledge.
Coment 5: The study applies Z-tests for proportion comparisons, but further explanation is needed on the rationale for choosing this test over other statistical approaches.
How were multiple comparisons controlled for? Include a justification for the statistical choices and, if applicable, whether corrections (e.g., Bonferroni) were used.
Response 5: We sincerely appreciate your careful review and valuable comments, which undoubtedly contribute to improving the quality and clarity of our work. We are expanding our response to address your concerns about statistical choices and provide a more comprehensive justification.
The argument for justifying using the Z-test for proportion comparisons and deciding to dichotomize the news tone variable is solid. To better justify this, we will add the following. The references have also been added to the bibliography.
Lines 275-281
The Z-test for proportion comparisons is a robust statistical tool for analyzing categorical or qualitative variables in large-sample studies. This method excels in determining significant differences between the proportions of two independent groups, making it invaluable for comparing distinct strategies or treatments [41]. The Z-test's strength lies in its capacity to provide meaningful insights into group differences while maintaining statistical rigor, especially when dealing with substantial sample sizes that ensure the validity of its underlying normal distribution approximation [42].
However, we acknowledge that it is necessary to consider and discuss the potential limitations of this approach. Thus, while recognising the potential for some nuance loss, we posit that the benefits of this approach in terms of analytical focus, statistical robustness, interpretability, and practical relevance outweigh its limitations in the context of our study. To better justify this, we will add the following:
Lines 258-271
While acknowledging that dichotomizing an originally polytomous variable may result in some information loss, our methodological decision to bifurcate "Tone of the News" categories into "Overall supporting " and "Other Tones" was predicated on carefully evaluating its merits and limitations. Although potentially simplifying the inherent complexity of tone variability in news, this approach offers several advantages that align with our research objectives and enhance the study's overall robustness. Primarily, it allows for a more focused analysis of positive tone news, which is central to our research aim.
Furthermore, this dichotomization yields larger sample sizes for each group, thereby increasing statistical power and enhancing the reliability of our results. The resulting binary classification also facilitates more precise interpretation and effective communication of findings, particularly to a broader audience. Moreover, in many practical contexts, this dichotomous distinction between clearly positive news and others may prove more actionable and valuable than a more granular categorization
Coment 6: Ensure that tables and figures are self-explanatory without requiring extensive text reference.
Response 6: Thank you very much for your suggestion. We have revised Table 2 by adding the dichotomized variables with several categories. Table 3 has been completely modified to make it more self-explanatory. The texts accompanying the tables have also been revised.
Lines 304-307
As mentioned in section 2.3, "Tone of News" has been dichotomized for proportion tests. So, the "Tone of the News" variable was split into "Overall Supportive" (57%) and "Other Tones" (43%). So, news projecting a positive image of DS for children was analyzed for proportional differences against other variables listed in the theoretical basis (Table 3).
Lines 314-324
Differences also emerged in the media used for publication (p=0.000). It is more likely to find news with an "Overall Supportive" tone in specialized print media (69%).
Similar results were found in intertextuality comparisons, showing significant differences (p=0.000), where news lacking personal or documentary sources had a more Overall Supportive tone (75% vs. 54% "Other Tones").
On the other hand, the differences in the proportions between the presence of "Appeal to consult a health care professional" and "DS mentioned in the headline" in the Tone are also statistically significant, although in a different direction to the previous variables mentioned: "Appeal to consult a health care professional" is more prevalent in the news with "Other Tones" (61.0%). Similarly, the "Appeal to consult a health care professional" is more likely in the news with "Other Tones" (42.4%).
Coment 7: The study currently classifies media as generalist vs. specialized, but further categorization (e.g., newspapers, blogs, social media, television) could provide deeper insights. A breakdown by media format and audience demographics.
Response 7: We appreciate your suggestion about the need for more detailed categorization of media and analysis of audience demographics. In our current study, we chose to simply categorize media as generalist or specialized due to the limitations of the data available through MyNews, which does not provide detailed information on the audience of each media medium.
However, we recognize the importance of understanding how different media formats (such as newspapers, blogs, social media, and television) and the demographic characteristics of their audiences can affect the reception and impact of information about dietary supplements for children. This approach would allow for a deeper understanding of variations in perception and communication effectiveness.
With this understanding, we consider including in future research a more comprehensive breakdown by media type and study of audience profiles. This would help us design communication strategies that better fit the needs and behaviors of different segments of the population, ensuring that key messages about children's health are more effective and accessible to all interested groups.
Lines 414-421
Analyzing the audiences and profiles of those who access news in different media would allow us to determine how information about DS for children is distributed and received in various media contexts. Understanding who the readers or viewers are and how they access this news will help us identify possible key differences in the exposure and interpretation of the content. This information is crucial to developing communication strategies that are effective and relevant to different segments of the population, ensuring that messages about children's health are adequately received and understood by various groups.
Coment 8: While the study references prior research, it does not sufficiently compare its findings to existing work on media framing of health topics. Discuss how results align or diverge from previous studies on dietary supplement communication.
Response 8: We thank the reviewer for pointing out the need to more thoroughly compare our findings with previous work on media framing of health issues. We have revised our discussion to include a more detailed comparison of how our results align with or diverge from previous studies on dietary supplement communication. Here we present the revised response:
Lines 334-369
Regarding the media's portrayal of DS for children, we observed a positive image that confirms its suitability to public opinion [23]. According to Scheufele and Tewksbury [25], this phenomenon is attributable to media framing, which often highlights the benefits of DS without adequately discussing the risks or the need for medical supervision. This tendency enhances the visibility and interest in DS and may promote a perception of necessity and efficacy not always supported by robust scientific evidence. Such framing may lead parents and caregivers to make potentially ill-informed decisions contrary to specialized medical recommendations.
Furthermore, the scarce discussion in the media about the risks associated with DS consumption may contribute to a lower perception of risk among the public. It is particularly worrying in the context of children's health, where supplementation decisions should be made with caution and based on professional advice.
Our findings align with trends observed in other studies on health communication. For example, studies such as those by Weeks [44] and Milazzo [45] have also reported a growth in media attention towards alternative health and supplementation issues, finding a predominantly positive tone in media coverage of alternative and complementary therapies. As in our study, while coverage is primarily positive, criticism about the effectiveness and safety of these interventions remains an underrepresented topic, which resonates with the observations of Caulfield et al. [33] on vitamin D supplementation, where the promotion of supplementation in the media without adequate discussion of its potential risks was highlighted.
A recent study [16] indicates that more than 75% of Spanish pediatricians recommend dietary supplements. This high recommendation rate could be reflected in the generally positive Tone found in our analysis of news articles, where 57% were "Generally in favor" of supplements. The frequent use of these products in daily pediatric practice may coincide with the positive media coverage observed in our study. This observation suggests that the positive representation of dietary supplements for children in the media could be influenced, in part, by standard prescribing practices among health professionals. However, it is relevant to consider that this correlation does not necessarily imply causation and that other factors could contribute to the supportive Tone in news coverage.
In this regard, news items showing a more positive stance often lacked documentary or personal sources to substantiate claims, warranting cautious interpretation. This is concerning as such media might promote economically driven information, especially significant given the recent sales increase in these products [2]. This finding is consistent with concerns expressed in a medical article [46] regarding the lack of child-specific evidence for many nutritional recommendations targeting young athletes.
Coment 9: The study assumes a direct influence of media on consumer perceptions without considering pre-existing biases (e.g., industry-sponsored media content, government health advisories). Introduce a discussion on commercial interests and potential conflicts of interest in media reporting.
Response 9: Our research assumes a direct influence of the media on consumer perceptions of dietary supplements for children. However, in the wake of your comment, we recognize limitations that underscore the need for future research addressing these areas for a more complete understanding of how media and commercial interests influence public perceptions and health decisions related to DS for children. Along these lines, we have introduced the following paragraph in the Discussion section:
Lines 429-437
Acknowledging that media reporting does not occur in a vacuum is essential. Commercial interests and potential conflicts of interest can significantly shape how dietary supplements are portrayed. To address this potential influence, future research could analyze media content sources, examine the relationship between regulatory changes and shifts in media portrayal of dietary supplements, and investigate how different stakeholders influence media narratives around children's dietary supplements. By incorporating these factors, we can provide a more comprehensive understanding of how the media shapes perceptions of dietary supplements for children, acknowledging the complex interplay of commercial interests, regulatory frameworks, and public health concerns
Coment 10: The conclusion hints at policy recommendations, but these should be more specific. Provide explicit recommendations for policymakers and journalists on improving public health communication regarding dietary supplements.
Response 10: We have rethought the text to introduce more explicit recommendations, and we have done so by introducing new proposals in the Discussion section.
Lines 438-460
The study provides some recommendations for media managers and policymakers. Given the risks of uncontrolled consumption of DS, the media should promote and take greater care of Intertextuality by using more credible sources. Contrasting It is advisable to contrast the sources used and make them explicit to evaluate their veracity and avoid leading people to make erroneous decisions. News about DS should consistently emphasize the need for parents and caregivers to consult healthcare providers before administering dietary supplements to children. It could be reinforced through expert quotes or case studies demonstrating the consequences of unsupervised supplement use. Journalists also should critically assess and challenge the widespread notion that "natural" products are inherently safe. This narrative can mislead consumers into thinking that supplements, due to their natural origin, are free from risks. Clear communication about the difference between "natural" and "safe" could help curb misconceptions.
On the other hand, the Public Administration should monitor compliance with the regulations, ensuring that the news does not have commercial interests without these being publicly known. Policymakers should fund research into dietary supplements' long-term effects and efficacy, particularly for children, to inform evidence-based health policies. It could include studies on potential interactions between supplements and medications commonly used by children. Government-led campaigns should focus on educating the public about the safe use of dietary supplements, emphasizing the importance of consulting healthcare professionals before using such products. These campaigns could follow successful strategies in food literacy with a multi-component approach, combining theoretical sessions with practical and interactive activities and relying on digital tools [53].
Coment 11: More innovative suggestions for future research are recommended. For example, in future research one could use natural language processing (NLP) and AI-driven sentiment analysis to assess bias and emotional tone in media reports on dietary supplements. Please see a recent research by Ravšelj et al., (2025) that might be useful by adding the most recent insights from the AI research.
Response 11:
Thank you very much for your comment. The text has been improved by incorporating references to methods for the analysis of emotional perceptions in media reports on dietary supplements. We thank you for the bibliographic reference on the use of AI research, which has also been incorporated in this work.
Lines 422-428
Likewise, to eliminate possible biases in evaluating the emotional Tone of news about DS, it would be advisable in future research to take advantage of advances in the field of Artificial Intelligence and Sensory Evaluation techniques. In this sense, the use of new techniques such as Natural Language Process (NLP), which would allow key information to be extracted from texts and detect emotions or feelings in comments, opinions, or social networks, or AI-driven sentiment analysis, would strengthen the evidence obtained [51, 52].

Reviewer 2 Report
Comments and Suggestions for Authors
This article investigates the media's influence on public perception of the safety of dietary supplements for children, highlighting how Spanish articles may reinforce a positive image of these products. Using a quantitative content analysis methodology to examine articles published between 2015 and 2021 provides a solid foundation for your findings, which indicate a 60% increase in publications promoting supplements, potentially leading to risky health behaviors and unsupervised consumption.
I suggest rephrasing some parts to increase clarity and impact. For example, the phrase regarding the lack of encouragement to consult medical professionals (Lines 23-24) could be modified to emphasize the importance of medical supervision in the consumption of supplements. Additionally, consider removing examples that deviate from the main focus of the analysis, such as the one related to minors with Down syndrome (Lines 141-144), which seems tangential to the main theme of the study.
Analyzing the use of data, the technique of comparing frequencies and proportions through the Z-test is well-executed, but could benefit from further details on levels of statistical significance and adjustments for multiple comparisons, if applicable.
The absence of a discussion on potential biases in media reporting, such as the influence of advertisers of supplement products, is a significant omission. Analyzing this aspect could further balance the results and provide a more comprehensive view of the media's role.
Finally, consider including the following citations to further enrich the discussion:
Mancone, S., et al. (2024). "Enhancing nutritional knowledge and self-regulation among adolescents: efficacy of a multifaceted food literacy intervention." Frontiers in psychology, 15, 1405414. Include it in the section discussing educational strategies for supplement use among adolescents.
This work is a thorough examination of a vital public health issue. With the suggested improvements and additional citations, it will undoubtedly contribute significantly to the discourse on dietary supplements and public health policy
Comments on the Quality of English LanguageThe quality of English in the manuscript is generally good, with clear and concise expressions consistent with academic standards. However, there are minor areas where language precision could be enhanced to better align with scholarly writing norms. For example, revising certain sentences to improve clarity and reduce ambiguity can further polish the presentation and ensure that the arguments are communicated effectively. Adding transitions between sections may also smooth the narrative flow, making the paper more reader-friendly. Overall, with these slight adjustments, the language quality will meet the high standards expected in international publications
Author Response
Coment 1: This article investigates the media's influence on public perception of the safety of dietary supplements for children, highlighting how Spanish articles may reinforce a positive image of these products. Using a quantitative content analysis methodology to examine articles published between 2015 and 2021 provides a solid foundation for your findings, which indicate a 60% increase in publications promoting supplements, potentially leading to risky health behaviors and unsupervised consumption
I suggest rephrasing some parts to increase clarity and impact. For example, the phrase regarding the lack of encouragement to consult medical professionals (Lines 23-24) could be modified to emphasize the importance of medical supervision in the consumption of supplements.
Response 1: We would like to thank the reviewer for their valuable comments and for the opportunity to review this manuscript. We appreciate their positive evaluation of the work and the suggestions for improving its quality. In pdf, we have highlighted our replies in blue and the changes incorporated into the manuscript in red.
Thank you very much for your comment 1. We are rephrasing some parts to increase clarity and impact.
Lines 14-37
Background/Objectives: The influence of media on public opinion, especially regarding health topics, is profound. This study investigates how Spanish media may reinforce a positive image of dietary supplements for children, potentially leading to harmful health attitudes and behaviors. Methods: The researchers conducted a quantitative content analysis of 912 news articles from Spanish media outlets discussing dietary supplements for children between 2015 and 2021. They used frequency analysis and proportion comparison to analyze variables such as the Reach of news, Tone of news, mentions of health professional consultation, association with natural products, media specialization, Intertextuality, and headline mentions. Results: The study found a 60% increase in publications discussing dietary supplements for children during the study period. The content analysis indicates that these articles predominantly present dietary supplements in a positive light promote a positive image of dietary supplements, often not backed by solid without robust scientific evidence. Moreover, Furthermore, many articles fail to encourage consultation with medical professionals, potentially leading to unsupervised consumption of these products among minors, do not emphasize the need for medical consultation, which may contribute to unsupervised consumption, particularly among minors. This highlights the critical importance of professional guidance when considering dietary supplements for children. Additionally, the frequent emphasis on the "natural" attributes of these products raises concerns regarding consumer perceptions and potential safety risks. Conclusions: The study reveals a concern problem regarding the portrayal of dietary supplements for children in Spanish media. The overly optimistic image, lack of scientific basis, and failure to recommend medical supervision may contribute to unsupervised consumption among minors, risking their health due to misinformed decisions influenced by media portrayal.
Coment 2: Additionally, consider removing examples that deviate from the main focus of the analysis, such as the one related to minors with Down syndrome (Lines 141-144), which seems tangential to the main theme of the study.
Response 2: The news item concerning Down's syndrome has been replaced. In its place, another example of a neutral tone has been added.
Lines 192-195
Provide comprehensive consumer information with specific labelling of food supplements, so that a medicinal plant cannot be sold as a supplement and a medicinal product at the same time. Establish maximum and minimum limits. In this way reference values will be well defined, especially for children.
Coment 3: Analyzing the use of data, the technique of comparing frequencies and proportions through the Z-test is well-executed, but could benefit from further details on levels of statistical significance and adjustments for multiple comparisons, if applicable.
Response 3: We sincerely appreciate your careful review and valuable comments, which undoubtedly contribute to improving the quality and clarity of our work. We are expanding our response to address your concerns about statistical choices and provide a more comprehensive justification.
We acknowledge that it is necessary to consider and discuss the potential limitations of this approach. Thus, while recognising the potential for some nuance loss, we posit that the benefits of this approach in terms of analytical focus, statistical robustness, interpretability, and practical relevance outweigh its limitations in the context of our study. To better justify this, we will add the following:
Lines 258-271
While acknowledging that dichotomizing an originally polytomous variable may result in some information loss, our methodological decision to bifurcate "Tone of the News" categories into "Overall supporting " and "Other Tones" was predicated on carefully evaluating its merits and limitations. Although potentially simplifying the inherent complexity of tone variability in news, this approach offers several advantages that align with our research objectives and enhance the study's overall robustness. Primarily, it allows for a more focused analysis of positive tone news, which is central to our research aim.
Furthermore, this dichotomization yields larger sample sizes for each group, thereby increasing statistical power and enhancing the reliability of our results. The resulting binary classification also facilitates more precise interpretation and effective communication of findings, particularly to a broader audience. Moreover, in many practical contexts, this dichotomous distinction between clearly positive news and others may prove more actionable and valuable than a more granular categorization
Coment 4: The absence of a discussion on potential biases in media reporting, such as the influence of advertisers of supplement products, is a significant omission. Analyzing this aspect could further balance the results and provide a more comprehensive view of the media's role.
Response 4: Our research assumes a direct influence of the media on consumer perceptions of dietary supplements for children. However, in the wake of your comment, we recognize limitations that underscore the need for future research addressing these areas for a more complete understanding of how media and commercial interests influence public perceptions and health decisions related to DS for children. Along these lines, we have introduced the following paragraph in the Discussion section:
Lines 429-437
Acknowledging that media reporting does not occur in a vacuum is essential. Commercial interests and potential conflicts of interest can significantly shape how dietary supplements are portrayed. To address this potential influence, future research could analyze media content sources, examine the relationship between regulatory changes and shifts in media portrayal of dietary supplements, and investigate how different stakeholders influence media narratives around children's dietary supplements. By incorporating these factors, we can provide a more comprehensive understanding of how the media shapes perceptions of dietary supplements for children, acknowledging the complex interplay of commercial interests, regulatory frameworks, and public health concerns.
Coment 5: Finally, consider including the following citations to further enrich the discussion:
Mancone, S., et al. (2024). "Enhancing nutritional knowledge and self-regulation among adolescents: efficacy of a multifaceted food literacy intervention." Frontiers in psychology, 15, 1405414. Include it in the section discussing educational strategies for supplement use among adolescents.
This work is a thorough examination of a vital public health issue. With the suggested improvements and additional citations, it will undoubtedly contribute significantly to the discourse on dietary supplements and public health policy
Response 5: Thank you for the reference, we learn about successful strategies in food literacy with a multi-component approach, combining theoretical sessions with practical and interactive activities and relying on digital tools. And we have introduced this learning as a specific recommendation for the designing of public campaigns.
Lines 457-459
These campaigns could follow successful strategies in food literacy with a multi-component approach, combining theoretical sessions with practical and interactive activities and relying on digital tools [53].
Coment 5: The quality of English in the manuscript is generally good, with clear and concise expressions consistent with academic standards. However, there are minor areas where language precision could be enhanced to better align with scholarly writing norms. For example, revising certain sentences to improve clarity and reduce ambiguity can further polish the presentation and ensure that the arguments are communicated effectively. Adding transitions between sections may also smooth the narrative flow, making the paper more reader-friendly. Overall, with these slight adjustments, the language quality will meet the high standards expected in international publications
Response 6: Thank you for your valuable feedback regarding the language quality. We have thoroughly reviewed the manuscript and addressed the suggestions provided. The language has been refined to improve clarity and reduce any ambiguity, ensuring that the arguments are communicated more effectively.

Round 2
Reviewer 1 Report
Comments and Suggestions for Authors
Authors have done well job on revising their manuscript.
Reviewer 2 Report
Comments and Suggestions for Authors
I am pleased to confirm that the manuscript has been revised successfully according to the specified guidelines. It is now in its final form and ready for publication.
Thank you to the authors for their diligent work and professionalism in addressing the revisions and meeting the required standards.
I kindly ask you to move forward with the final stages of the publication process.
Comments on the Quality of English LanguageThe manuscript has been reviewed and the quality of the English language is commendable. The authors have demonstrated a clear and concise use of language that supports the effective communication of their ideas. There are minor areas where further refinements could enhance clarity and flow, but overall, the manuscript meets high linguistic standards suitable for publication.
Thank you for your attention to maintaining a high level of clarity and professionalism in your writing.